# PRAME Immunocytochemistry for the Diagnosis of Melanoma Metastases in Cytological Samples

**DOI:** 10.3390/diagnostics12030646

**Published:** 2022-03-06

**Authors:** Andrea Ronchi, Federica Zito Marino, Elvira Moscarella, Gabriella Brancaccio, Giuseppe Argenziano, Teresa Troiani, Stefania Napolitano, Renato Franco, Immacolata Cozzolino

**Affiliations:** 1Pathology Unit, Department of Mental and Physical Health and Preventive Medicine, Università degli Studi della Campania “Luigi Vanvitelli”, Via Luciano Armanni 5, 80100 Naples, Italy; andrea.ronchi@unicampania.it (A.R.); federica.zitomarino@unicampania.it (F.Z.M.); coimma73@gmail.com (I.C.); 2Dermatology Unit, Department of Mental and Physical Health and Preventive Medicine, Università degli Studi della Campania “Luigi Vanvitelli”, Via Luciano Armanni 5, 80100 Naples, Italy; elvira.moscarella@unicampania.it (E.M.); gabriella.brancaccio@unicampania.it (G.B.); giuseppe.argenziano@unicampania.it (G.A.); 3Oncology Unit, Department of Precision Medicine, Università degli Studi della Campania “Luigi Vanvitelli”, Via Luciano Armanni 5, 80100 Naples, Italy; teresa.troiani@unicampania.it (T.T.); stefania.napolitano@unicampania.it (S.N.)

**Keywords:** melanoma metastasis, immunocytochemistry, SOX10, S100, Melan-A, HMB45, PRAME

## Abstract

(1) Background: Fine-needle aspiration cytology is often used for the pre-operative diagnosis of melanoma metastases. The diagnosis may not be confidently established based on morphology alone, and immunocytochemistry is mandatory. The choice of the most advantageous immunocytochemical antibodies is critical, as the sample may be scant, and the presence of pigmented histiocytes may be confounding. However, the diagnostic performance of melanocytic markers in this setting is poorly investigated. Moreover, PRAME (preferentially expressed antigen in melanoma) recently emerged as a novel marker for the diagnosis of melanoma. The current work aimed to evaluate the sensitivity and specificity of PRAME for the diagnosis of melanoma metastases in cytological samples, compared to other melanocytic markers. (2) Methods: PRAME, S100, Melan-A, HMB45 and SOX10 were tested on cell block sections of 48 cases of melanoma metastases diagnosed from cytological samples, and 20 cases of reactive lymphadenopathy. (3) Results: S100 and SOX10 showed the highest sensitivity (100%), while the sensitivity of PRAME was 85.4%. PRAME, Melan-A, SOX10 and HMB45 showed a specificity of 100%, while the specificity of S100 was lower (85%), as it marked some histiocytes. (4) Conclusion: PRAME immunocytochemistry is highly specific for the diagnosis of melanoma metastasis from a cytological sample, but is less sensitive compared with other melanocytic markers.

## 1. Introduction

Cutaneous melanoma (CM) is an aggressive neoplasm with a high rate of metastasis, depending on the stage of the neoplasm and the Breslow thickness [1]. Lymph node metastases are relatively frequent in patients affected by CM, and direct sampling is mandatory to obtain a correct diagnosis. Fine-needle aspiration cytology (FNAC) is recommended by relevant guidelines as a useful tool to obtain a direct sample of the neoplasm, allowing a diagnosis of CM metastasis [2]. Moreover, FNAC is a useful tool to obtain neoplastic cells to perform molecular evaluations, for a predictive purpose, on metastatic neoplasms [3]. The diagnosis of CM metastasis from a cytological sample may be challenging, as the morphology of the neoplastic cells is widely variable, including several types, such as epithelioid, plasmacytoid, spindle, small, rhabdoid, signet-ring, myxoid and balloon cells [4]. In this setting, immunocytochemistry (ICC) plays a crucial role in diagnosis, when morphology alone has insufficient diagnostic accuracy [5]. Nevertheless, the diagnostic performance of the different ICC antibodies applicable to cytological samples for the diagnosis of CM metastases is poorly investigated. SOX10 and S100 have higher sensitivity (100%), while Melan-A and HMB45 showed sensitivities of 97% and 95%, respectively [6]. PRAME (preferentially expressed antigen in melanoma) is a tumor-associated antigen expressed by some neoplasms, which recently emerged as a novel immunohistochemical marker for the diagnosis of CM [7,8,9]. Although some evidence demonstrated that PRAME is expressed in most primary and metastatic CM, data about the diagnostic performance of PRAME for the diagnosis of CM metastases in cytological samples are missing.

The aim of this study was to investigate the diagnostic performance of PRAME for the diagnosis of CM metastasis in a series of FNAC samples, compared to the other ICC melanocytic markers.

## 2. Materials and Methods

### 2.1. Case Selection

All cases with a previous diagnosis of CM, which FNAC had subsequently been performed on, between January 2017 and September 2021, were retrieved from the archives of the Pathology Unit of “Vanvitelli” University in Naples. The inclusion criteria were the following: (1) the diagnosis was rendered based on the cytological sample; (2) a cell block (CB) was available; (3) the lesion was subsequently excised and the diagnosis was confirmed histologically. Sixty-eight cases met the inclusion criteria, including 48 cases of CM metastases and 20 cases of reactive lymphadenopathies.

### 2.2. Sample Management

FNAC was performed in all cases by a cytopathologist by ultrasound (US) or computer tomography (CT) guidance. Air-dried and alcohol-fixed smears were prepared in all cases and stained by Diff-Quick and Papanicolaou methods, respectively. A dedicated pass was executed in all cases after rapid on-site evaluation and was suspended in 5 mL formalin for the realization of the CB. Formalin-fixed and paraffin-embedded CB sections were used for the ICC evaluations. A CB section was stained with haematoxylin and eosin for adequacy and morphological evaluation of the cytological material.

### 2.3. Immunocytochemistry

Immunocytochemistry was performed on consecutive CB sections. It was performed on a Ventana platform (Ventana BenchMark ULTRA system) according to the manufacturer’s instructions, using the following antibodies: Melan-A (mouse monoclonal primary antibody, clone A103), HMB45 (mouse monoclonal primary antibody, clone HMB45), S100 (mouse monoclonal primary antibody, clone 4C4.9), SOX10 (rabbit monoclonal primary antibody, clone SP267) and PRAME (rabbit monoclonal antibody, clone EPR20330). Immunocytochemistry was interpreted according to the scoring system previously described, considering both the percentage of neoplastic stained cells (score 0–3) and the staining intensity (score 0–3) [6]. The final ICC score was derived from the sum of the two individual scores, as follows: 0: negative expression; 1–3: weak expression; 4: moderate expression; 5–6: strong expression. In particular, the percentage of positive cells was evaluated as follows: 0 (no positive cells); 1 (≤10% positive cells); 2 (10–50% positive cells); 3 (>50% positive cells). Staining intensity was evaluated as follows: 0 (no positivity); 1 (barely perceptible positivity); 2 (distinctly recognizable positivity); 3 (intense positivity). The final ICC score was derived from the sum of the two individual scores, as follows: 0: negative expression; 1–3: weak expression; 4: moderate expression; 5–6: strong expression.

### 2.4. Statistical Analysis

The sensitivity was calculated for each ICC antibody. Furthermore, a two-sided Wilcoxon signed-rank test was used for the comparison of the diagnostic performances of the two most sensitive antibodies (SOX10 and S100). The test compared the final PRAME ICC scores with the final ICC scores of SOX10 and S100 for the 48 cases of CM metastases. The test was considered statistically significant for *p* < 0.05. The statistical analysis was carried out using IBM SPSS statistics V.20.

### 2.5. Ethical Consideration

The present study was retrospectively conducted on archival biological samples. The cytological diagnoses and their histological confirmations had already been rendered in all cases. At the time of the FNAC procedure, written consent, including consent to use the diagnostic data for scientific purposes, had been obtained from each patient. Approval by the institutional ethics board was attained.

## 3. Results

Our series included 48 cases of CM metastases diagnosed from cytological samples. The diagnosis was subsequently confirmed histologically in all cases. The metastases were in the lymph nodes in 44 (91.6%) cases, in subcutaneous tissues in 2 (4.2%) cases and in the lungs in 2 (4.2%) cases.

PRAME was positive in 41 (85.4%) cases and negative in the remaining 7 (14.6%) cases. Strong positivity was observed in 38 out of 41 (92.7%) positive cases, and weak positivity was observed in the remaining 3 out of 41 (7.3%) positive cases. The sensitivity of PRAME ICC was 85.4%.

S100 was positive in all 48 cases (100%). Strong positivity was observed in 43 out of 48 (89.6%) cases, moderate positivity was observed in 3 out of 48 (6.2%) cases, and weak positivity was observed in the remaining 2 (4.2%) cases. The sensitivity of S100 ICC was 100%.

SOX10 was positive in all 48 cases (100%). Forty-six out of 48 (96%) cases showed strong positivity, 1 out of 48 (2%) cases showed moderate positivity, and the remaining 1 (2%) case showed weak positivity. The sensitivity of SOX10 was 100%.

Melan-A was positive in 47 out of 48 (97.9%) cases. One out of 48 (2%) cases was negative. Strong positivity was observed in 23 out of 47 (48.9%) positive cases, moderate positivity in 18 out of 47 (38.3%) cases, and weak positivity in 6 (12.8%) cases. The sensitivity of Melan-A was 97.9%.

HMB45 was positive in 43 out of 48 (89.6%) cases. Five out of 48 (10.4%) cases were negative. Twenty-seven out of 43 (62.8%) positive cases showed strong positivity, 11 out of 43 (25.6%) positive cases showed moderate positivity, and the remaining 5 (11.6%) positive cases showed weak positivity. The sensitivity of HMB45 was 89.6%.

The statistical analysis demonstrated a significantly lower diagnostic performance of PRAME compared with S100 (*p* = 0.0004) and SOX10 (*p* = 0.00001).

Our series also included 20 cases of reactive lymphadenopathy diagnosed from cytological samples, and subsequently confirmed histologically. In these cases, PRAME, SOX10, Melan-A and HMB45 were negative in 20 out of 20 cases (100%), while S100 marked histiocytes in 3 out of 20 (15%) cases, making the distinction between histiocytes and melanocytes difficult. The specificity of PRAME, SOX10, Melan-A and HMB45 was 100%, while the specificity of S100 was 85%.

The results are summarized in Table 1 and Figure 1. Some examples of the ICC staining are showed in Figure 2.

## 4. Discussion

Fine-needle aspiration cytology is widely applied for the diagnosis of CM metastases, mainly in patients with a known history of CM. However, the diagnosis of CM from cytological samples may be challenging, as CM may present heterogeneous morphological findings [4]. Moreover, pigmented histiocytes may be present in the sample, and may be confused with neoplastic cells [4]. Nevertheless, a correct diagnosis of CM metastases is mandatory, because cytological samples may also be used for the predictive evaluation of the BRAF molecular status of the neoplasm, and because immunotherapy has recently been introduced for patients affected by metastatic CM [3]. In this setting, ICC plays a crucial role in simultaneously obtaining the best diagnostic yield and optimizing sample handling. As neoplastic cells may be scant in a cytological sample, it is important to choose the best ICC markers. PRAME has recently emerged as a novel immunohistochemical marker useful for the differential diagnosis of melanocytic neoplasms [9,10,11]. Lezcano et al. have recently demonstrated that up to 94% of CMs show diffuse immunopositivity for PRAME, while benign nevi were negative, or only focally positive [12]. However, the diagnostic performance of PRAME for the diagnosis of CM is still poorly defined, and it has not been previously indagated in cytological samples. In this study, we investigated the sensitivity and specificity of PRAME in a cytological series, which included 48 CM metastases and 20 reactive lymphadenopathies of patients with previous CM diagnosis, comparing the sensitivity and specificity of PRAME with other melanocytic markers. In a previous study, we demonstrated that S100 and SOX10 are the most sensitive markers [6]. Herein, we confirmed that SOX10 and S100 have the highest sensitivity for the diagnosis of CM in cytological samples from CM metastases, demonstrating a sensitivity of 100% for both the ICC antibodies. SOX10 was the most useful ICC marker, as SOX10 staining showed strong positivity in 96% of cases, while S100 staining showed strong positivity in 89.6% of cases. Moreover, the nuclear staining of SOX10 was easier to detect, not allowing confusion with melanin pigment or pigmented histiocytes. Melan-A, HMB45 and PRAME ICC antibodies showed sensitivities of 97.9%, 89.6% and 85.4%, respectively. Most cases (38) showed strong positivity for PRAME, while only three cases showed weak positivity. Nevertheless, PRAME demonstrated the lowest sensitivity. In reactive lymphadenopathies cases diagnosed on cytological samples, PRAME, SOX10, Melan-A and HMB45 were negative in all cases, resulting in a specificity of 100%. On the other hand, S100 marked the histiocytes in 15% of cases, resulting in a specificity of 85%. S100 positivity in histiocytes may be challenging, as they may be misinterpreted as malignant cells, mainly when pigmented histiocytes are present in haematoxylin and eosin-stained sections (Figure 3). In this study, PRAME demonstrated the optimal specificity (100%), but not high sensitivity (85.4%). In particular, PRAME showed the lowest sensitivity compared to the other melanocytic ICC markers. These data are not surprising because the importance of PRAME in the diagnosis of CM is primarily based on its high specificity, and not on its sensitivity [12]. It is difficult to make assumptions about the reasons for this poor sensitivity. Indeed, the function of PRAME in CM remains largely unknown. Epping et al. have characterized PRAME as a ligand-dependent co-repressor of retinoic acid receptor α (RARα), RARβ and RARγ signaling [13]. Although data suggest that PRAME is expressed by a large percentage of CM, and not by benign melanocytic nevi, the exact role of PRAME in the CM molecular landscape is unknown. Based on the actual data, it is reasonable to assume that PRAME regulation in CM cells is complex, and is mainly due to epigenetic mechanisms [14]. Consequently, not all CMs express PRAME, and so its sensitivity is lower than other melanocytic ICC markers. On the other hand, SOX10 is a nuclear transcription factor that plays an important role in melanocytic cell differentiation [15]. It is, therefore, not surprising that SOX10 is expressed by most CMs. Obviously, SOX10 is useless for the differential diagnosis of primary cutaneous lesions between CM and benign melanocytic nevi, while PRAME is more informative in this diagnostic setting. However, in the setting of the diagnostic evaluation of cytological samples in cases of suspected CM metastases, SOX10 remains the most sensitive and useful ICC marker.

## 5. Conclusions

ICC is mandatory in the evaluation of cytological samples for the diagnosis of CM metastasis. PRAME showed low sensitivity when compared with other melanocytic markers, but showed high specificity. SOX10 and S100 are confirmed as the most sensitive markers in this specific setting, but S100 was less specific. In conclusion, SOX10 is the most useful ICC marker for the diagnosis of CM metastasis from cytological samples. PRAME is not a suitable ICC marker in this setting, although some data suggest that PRAME may play an important role in the diagnosis of CM in cutaneous samples.

## Figures and Tables

**Figure 1 diagnostics-12-00646-f001:**
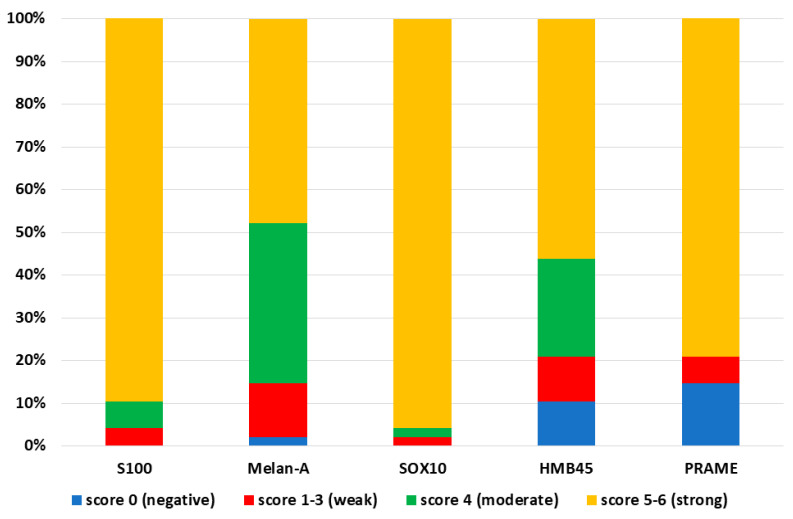
Distribution of staining scores for the immunocytochemistry antibodies. Immunocytochemical stains for S100, Melan-A, SOX10, HMB45 and PRAME were performed in 48 cytological samples of melanoma metastases. S100 and SOX10 were positive in all cases (100%). Melan-A, HMB45 and PRAME were negative in 1 (2.1%), 5 (10.4%) and 7 (14.6%) cases, respectively. SOX10 showed the best diagnostic performance, as strong positivity was observed in 46 (95.8%) cases.

**Figure 2 diagnostics-12-00646-f002:**
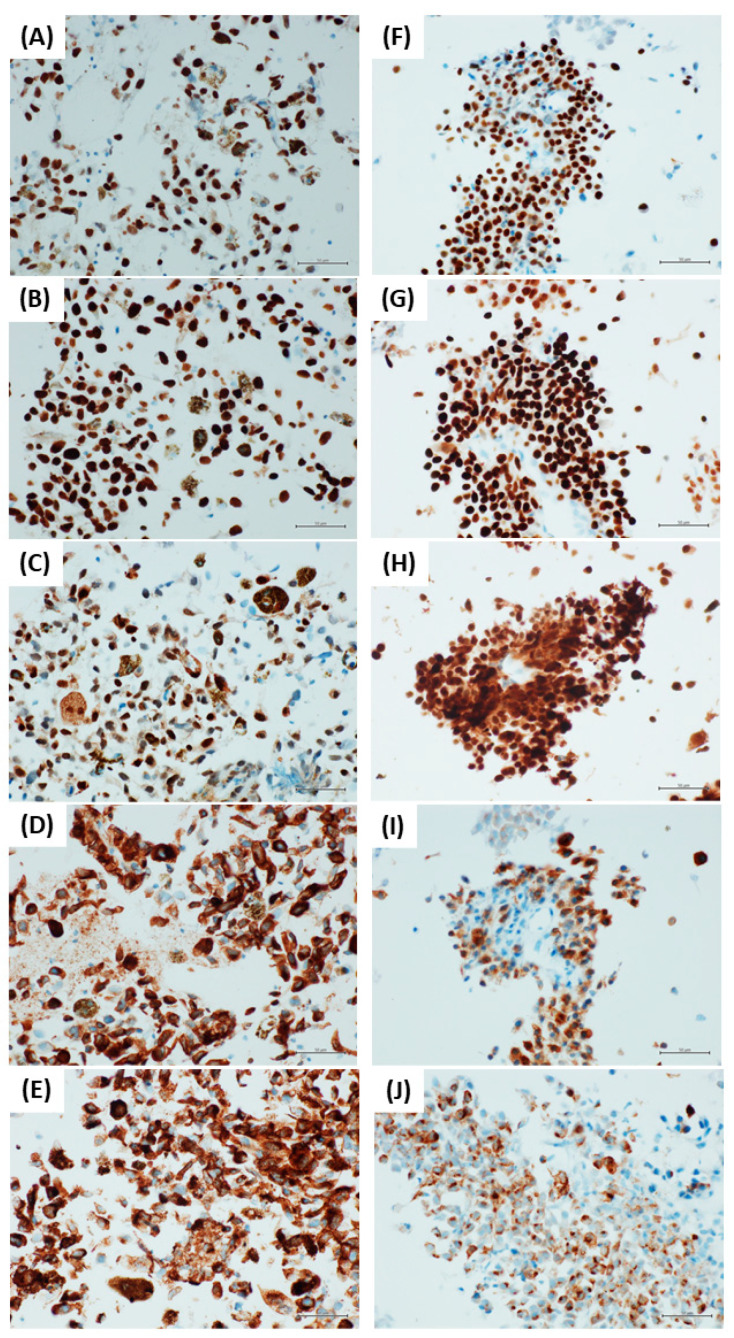
Immunohistochemistry in a case of heavily pigmented melanoma metastasis (**A**–**E**), and in a case of slightly pigmented melanoma metastasis (**F**–**J**) ((**A**,**F**): PRAME, (**B**,**G**): SOX10, (**C**,**H**): S100, (**D**,**I**): Melan-A, and (**E**,**J**): HMB45). ((**A**–**J**): immunostain, original magnification 400×).

**Figure 3 diagnostics-12-00646-f003:**
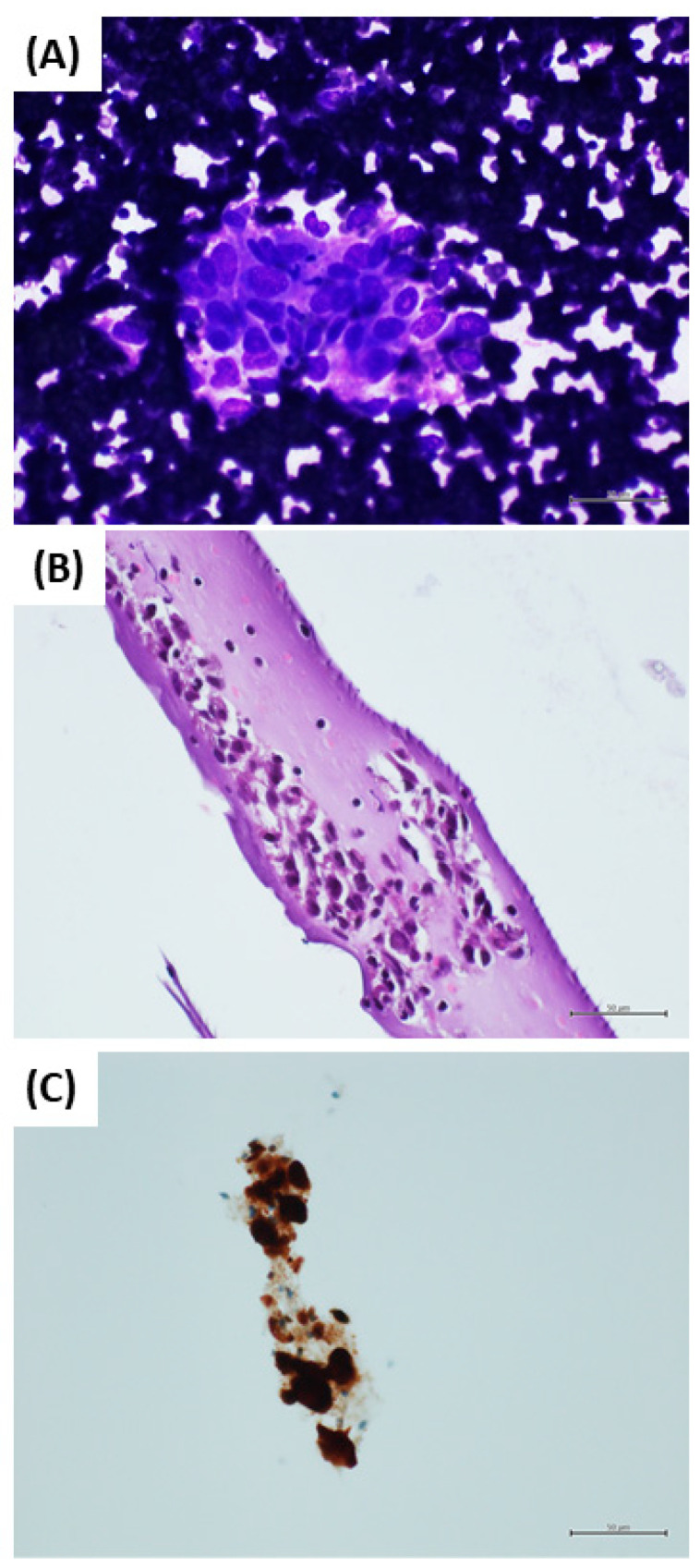
Inguinal lymph node fine-needle aspiration cytology: (**A**) direct smear: cell aggregate consisting of medium–large-sized cells with irregular nuclear membrane and large eosinophilic cytoplasm. In this context, only a few small lymphocytes are recognized. Morphologically, these aspects are not clearly identifiable with a macrophage–lymphocyte complex (May Grunwald Giemsa stain, 400×). (**B**) Cell block section: medium-sized cells with round–oval nuclei, sometimes incised and granular cytoplasm, eosinophilic. Small, mature lymphocytes are also present (hematoxylin and eosin stain, original magnification 400×). (**C**) S100 immunostaining in this cell aggregate is positive; PRAME and SOX10 are negative (S100 immunostain, original magnification 400×).

**Table 1 diagnostics-12-00646-t001:** Diagnostic performance of immunocytochemical markers.

	PRAME	S100	SOX10	Melan-A	HMB45
Positive (N. cases)	41	48	48	47	43
Negative (N. cases)	7	0	0	1	5
Strong positivity (N. cases)	38	43	46	23	27
Moderate positivity (N. cases)	0	3	1	18	11
Weak positivity (N. cases)	3	2	1	6	5
Sensitivity	85.4%	100%	100%	97.9%	89.6%
Specificity	100%	85%	100%	100%	100%

## Data Availability

Data are available upon reasonable request. All data relevant to the study are included in the article. For additional information please contact corresponding author.

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
