# Peer review of "PRAME Immunocytochemistry for the Diagnosis of Melanoma Metastases in Cytological Samples"

_diagnostics, 2022, doi:10.3390/diagnostics12030646_

Round 1

Reviewer 1 Report

Diagnostics (ISSN 2075-4418)

Manuscript ID: diagnostics-1611459

The MS entitled PRAME immunocytochemistry for the diagnosis of melanoma metastases on cytological samples. by Ronchi et al., is a valuable study dedicated to the new cytological and immunocytochemical methods applications for the identification of the melanoma cells. The advances of the new and existing methods are evaluated. A panel of several antibodies for the diagnostic was analyzed. The MS can be valuable for the specialists in the area of cancer researches. The data presented are of important scientific significance.  So in my opinion the MS can be published with some changes performed.

An abstract and introduction are clearly written but there are concern about MS. At once it was not absolutely clear to me the method used and the table and diagram presentation. I would of recommend to add some more information on it to make it more understandable.

The MS itself is quite short for my opinion and can be considered as a short note in its present view.

I also would like that author can answer of some question in the text of MS.

From methods section it is unclear where it was a consecutive staining of samples by different abs?

Do cases mean histological samples - not clear how many sample were staining for the data presenting on histograms? numbers of what?

 there is also some misprint - line 166.

 What number means in table?

Why you gave different magnification in these two columns on fig.2? It would more reasonable (for me) to have the same magnification - to compare.

Conclusion(s) are short, so it is a conclusion. Scale bars on figs are hardly recognizable. The quality of photos is good.

How the statistical analysis was performed – it is unclear what was compared with what, ns a just numbers ( or %) for some cases looks very  close for statistics?

How many replicas were used (for statistic?) and even not for statistic – I am interesting not in number of cases but in number of sample studies < slides studied for every case?

 How the staining intensity was measured??

 How you estimate the sensitivity of staining?

 It is mentioned that you have scored the number of cells stained – may be it will be useful to introduce these data in MS?

As a report is short – I advice to explain it more wildly – it will make stronger your results and conclusions.

You might also widen your conclusion.

The discussion section is also quite short and mostly repeated the results, I would of like to hear there some comments on the sensitivity of staining ( reasons) and specificity of staining in terms of some molecular evidences or so.

line 98 – 99: style – interpreted, interpretation

Author Response

Renato Franco MD PhD

Full Professor

Pathology Unit

Università degli Studi della Campania “L. Vanvitelli”

Via Luciano Armanni 20

80100 Naples

email: [email protected]

Manuscript ID: diagnostics-1611459

Title: PRAME immunocytochemistry for the diagnosis of melanoma metastases on cytological samples.

Dear Reviewer,

Thank you for your precious observations, based on which we made the following changes:

REVIEWER 1

The MS (MANUSCRIPT) entitled PRAME immunocytochemistry for the diagnosis of melanoma metastases on cytological samples. by Ronchi et al., is a valuable study dedicated to the new cytological and immunocytochemical methods applications for the identification of the melanoma cells. The advances of the new and existing methods are evaluated. A panel of several antibodies for the diagnostic was analyzed. The MS can be valuable for the specialists in the area of cancer researches. The data presented are of important scientific significance.  So in my opinion the MS can be published with some changes performed.

AA: Thank you for your interest in our work.

An abstract and introduction are clearly written but there are concern about MS. At once it was not absolutely clear to me the method used and the table and diagram presentation. I would of recommend to add some more information on it to make it more understandable.

The MS itself is quite short for my opinion and can be considered as a short note in its present view.

AA: According to your suggestions, we have expanded the manuscript, by adding further considerations in the Discussion section, about the reasons of differences in sensitivity of the staining and potential molecular explanations. We stated more clearly the methods, particularly for immunocytochemistry performance and assessment.

I also would like that author can answer of some question in the text of MS.

From methods section it is unclear where it was a consecutive staining of samples by different abs?

AA: The 5 immunocytochemical antibodies were performed on consecutive cell-block sections. We specified it in the text (Methods section).

Do cases mean histological samples - not clear how many sample were staining for the data presenting on histograms? numbers of what?

AA: Histograms represent the diagnostic performance of each immunocytochemistry antibody. Immunocytochemistry was performed in 48 cytological samples of melanoma metastases. We have expanded the legend of the Figure 1, to make it clearer.

there is also some misprint - line 166.

AA: We corrected it.

What number means in table?

AA: Number in table represent number of cases (positive cases, negative cases, etc). We specified it in the table.

Why you gave different magnification in these two columns on fig.2? It would more reasonable (for me) to have the same magnification - to compare.

AA: Thank you for your attention. Pictures in Figure 2 are all at the same magnification (400x), indeed. The legend in the previous manuscript was indeed wrong. We corrected the legend of the figure.

Conclusion(s) are short, so it is a conclusion.

AA: We corrected the title of the section, that is now named “conclusion”.

Scale bars on figs are hardly recognizable. The quality of photos is good.

AA: Thank you for your appreciation.

How the statistical analysis was performed – it is unclear what was compared with what, ns a just numbers (or %) for some cases looks very close for statistics?

AA: Wilcoxon signed-rank test was used for the comparison of the diagnostic performances of PRAME compared to the two most sensitive antibodies, as SOX10 and S100. The test compared the PRAME final ICC scores with SOX10 and S100 final ICC scores, for the 48 cytological cases of melanoma metastases. We clarified it in the Methods section.

How many replicas were used (for statistic?) and even not for statistic – I am interesting not in number of cases but in number of sample studies < slides studied for every case?

AA: Each ICC antibody was performed on a single cell-block section. Every section included a positive-control. As all the samples were adequately cellular and representative, a single section was sufficient.

How the staining intensity was measured?

AA: Staining intensity was assessed manually, as it is actually assessed routinary in everyday pathologists’ practice. The ICC score we used includes both intensity of the staining and percentage of positive cells. The score system was previously used for the evaluation of melanocytic ICC markers on cytological samples.

How you estimate the sensitivity of staining?

AA: The Series is composed by 48 cases of melanoma metastases and 20 cases of reactive lymphadenopathy diagnosed on cytological sample and subsequently confirmed histologically. So, the series includes 48 true positive cases and 20 true negative cases. Sensitivity of each ICC staining was calculated: N true positives / (N true positives + N false negatives).

It is mentioned that you have scored the number of cells stained – may be it will be useful to introduce these data in MS?

AA: We stated more clearly the immunocytochemical scoring system in the Methods section. This scoring system has already been used for the evaluation of the diagnostic performance of immunocytochemical markers. The final scores of each antibody are stated in the text (Results section) and showed in the Figure 1 for a faster evaluation.

As a report is short – I advice to explain it more wildly – it will make stronger your results and conclusions.

You might also widen your conclusion.

The discussion section is also quite short and mostly repeated the results, I would of like to hear there some comments on the sensitivity of staining ( reasons) and specificity of staining in terms of some molecular evidences or so.

AA: We added further considerations in the Discussion section, about the reasons of differences in sensitivity of the staining and potential molecular explanations.

line 98 – 99: style – interpreted, interpretation

AA: We corrected it.

All the changes in the manuscript are marked in red.

Best regards,

                                                                                                Renato Franco

Reviewer 2 Report

With respect to sensitivity and specificity, SOX10 is superior to all other markers. It could be helpful to point this out even more clearly in the abstract and/or the conclusion. Is there a setting in which I would choose PRAME instead of SOX10 or would you reccommend using both?

Potentially rephrase:

"with high rate of metastases" -> "with a high rate of metastasis" or "with high rates of metastases" ?
Line 57 "neoplastic cell" -> "neoplastic cells"

"Furthermore, Wilcoxon signed-rank test was used for the comparation of the diagnostic performances of the two most sensitive antibodies, as SOX10 and S100."
-> "Furthermore, a two-sided? Wilcoxon signed-rank test was used for the comparison of the diagnostic performances of the two most sensitive antibodies, as SOX10 and S100."

Line 130  "One out of 48 (2%) case" -> " One out of 48 (2%) cases"

Line 134 "Five out of 48 (%) cases resulted" is unclear without the percentage given.

Line 139, please always give the full actual p-value as a number and do not use binarization of p-values.

Line 143 ", making difficult the distinction between histiocytes and melanocytes." -> ", making the distinction between histiocytes and melanocytes difficult."

As there are some typos and grammar (plural/singluar) issues I'd suggest to have (the coauthors) or other persons proof-read the manuscripts thoroughly before submission. Some of these typos should have been visible in a simple google drive or word document check.

Line 159 "et al" -> "et al. <italic>"

Line 166 " n a previous study"  - typo

Author Response

Renato Franco MD PhD

Full Professor

Pathology Unit

Università degli Studi della Campania “L. Vanvitelli”

Via Luciano Armanni 20

80100 Naples

email: [email protected]

Manuscript ID: diagnostics-1611459

Title: PRAME immunocytochemistry for the diagnosis of melanoma metastases on cytological samples.

Dear reviewer,

Thank you for your precious observations, based on which we made the following changes:

REVIEWER 2

With respect to sensitivity and specificity, SOX10 is superior to all other markers. It could be helpful to point this out even more clearly in the abstract and/or the conclusion. Is there a setting in which I would choose PRAME instead of SOX10 or would you reccommend using both?

AA: We stated more clearly in the conclusion that SOX10 is the most useful marker. In our opinion and according to our data, we prefer to use SOX10 for the diagnosis of melanoma metastasis on cytological samples, while PRAME seems not so useful in this specific setting.

Potentially rephrase:

"with high rate of metastases" -> "with a high rate of metastasis" or "with high rates of metastases" ?

Line 57 "neoplastic cell" -> "neoplastic cells"

"Furthermore, Wilcoxon signed-rank test was used for the comparation of the diagnostic performances of the two most sensitive antibodies, as SOX10 and S100."

-> "Furthermore, a two-sided? Wilcoxon signed-rank test was used for the comparison of the diagnostic performances of the two most sensitive antibodies, as SOX10 and S100."

Line 130  "One out of 48 (2%) case" -> " One out of 48 (2%) cases"

Line 134 "Five out of 48 (%) cases resulted" is unclear without the percentage given.

AA: Thank you for your attention. We corrected the typos.

Line 139, please always give the full actual p-value as a number and do not use binarization of p-values.

AA: We gave the full values.

Line 143 ", making difficult the distinction between histiocytes and melanocytes." -> ", making the distinction between histiocytes and melanocytes difficult."

AA: We corrected the sentence.

As there are some typos and grammar (plural/singluar) issues I'd suggest to have (the coauthors) or other persons proof-read the manuscripts thoroughly before submission. Some of these typos should have been visible in a simple google drive or word document check.

AA: Thank you for your attention. We proof-read the manuscript thoroughly and corrected the typos.

Line 159 "et al" -> "et al. <italic>"

Line 166 " n a previous study"  - typo

AA: We corrected the typos.

All the changes in the manuscript are marked in red.

Best regards,

                                                                                                Renato Franco
